# Metaproteomics Approach and Pathway Modulation in Obesity and Diabetes: A Narrative Review

**DOI:** 10.3390/nu14010047

**Published:** 2021-12-23

**Authors:** Francesco Maria Calabrese, Annalisa Porrelli, Mirco Vacca, Blandine Comte, Katharina Nimptsch, Mariona Pinart, Tobias Pischon, Estelle Pujos-Guillot, Maria De Angelis

**Affiliations:** 1Department of Soil, Plant and Food Science, Aldo Moro University, Bari, Via G. Amendola 165/a, 70126 Bari, Italy; a.porrelli5@studenti.uniba.it (A.P.); mirco.vacca@uniba.it (M.V.); maria.deangelis@uniba.it (M.D.A.); 2INRAE, UNH, Metabolism Exploration Platform, MetaboHUB Clermont, Clermont Auvergne University, F-63000 Clermont-Ferrand, France; Blandine.Comte@inrae.fr (B.C.); estelle.pujos-guillot@inrae.fr (E.P.-G.); 3Max Delbrück Center for Molecular Medicine in the Helmholtz Association (MDC), Molecular Epidemiology Research Group, 13125 Berlin, Germany; Katharina.Nimptsch@mdc-berlin.de (K.N.); Mariona.PinartGilberga@mdc-berlin.de (M.P.); tobias.pischon@mdc-berlin.de (T.P.); 4Charité-Universitätsmedizin Berlin, Corporate Member of Freie Universität Berlin, Humboldt-Universität zu Berlin, 10117 Berlin, Germany; 5German Centre for Cardiovascular Research (DZHK), Partner Site Berlin, 10785 Berlin, Germany; 6Biobank Technology Platform, Max Delbrück Center for Molecular Medicine in the Helmholtz Association (MDC), 13125 Berlin, Germany; 7Biobank Core Facility, Berlin Institute of Health at Charité-Universitätsmedizin Berlin, 10178 Berlin, Germany

**Keywords:** metaproteomics, low-grade inflammation, obesity, diabetes, gut microbiota, metabolic diseases

## Abstract

Low-grade inflammatory diseases revealed metabolic perturbations that have been linked to various phenotypes, including gut microbiota dysbiosis. In the last decade, metaproteomics has been used to investigate protein composition profiles at specific steps and in specific healthy/pathologic conditions. We applied a rigorous protocol that relied on PRISMA guidelines and filtering criteria to obtain an exhaustive study selection that finally resulted in a group of 10 studies, based on metaproteomics and that aim at investigating obesity and diabetes. This batch of studies was used to discuss specific microbial and human metaproteome alterations and metabolic patterns in subjects affected by diabetes (T1D and T2D) and obesity. We provided the main up- and down-regulated protein patterns in the inspected pathologies. Despite the available results, the evident paucity of metaproteomic data is to be considered as a limiting factor in drawing objective considerations. To date, ad hoc prepared metaproteomic databases collecting pathologic data and related metadata, together with standardized analysis protocols, are required to increase our knowledge on these widespread pathologies.

## 1. Introduction

Gut microbiota modulates the innate and adaptive immune systems both locally in the intestinal mucosa and outside the gut. Variations in microbial pattern abundance have been associated with certain autoimmune or inflammatory diseases known as ‘metabolic diseases’. In this field, type 1 and type 2 diabetes (T1D and T2D, respectively) and obesity are the most common and prevalent diseases featured by metabolic perturbations also involving gut microbiota dysbiosis [1,2]. Evidence-based data revealed how changes in gut microbiome contribute to an increased susceptibility to the onset and development of several diseases [3]. The main actors of these mechanisms are the colonic microbiota, its metabolic products, and the host immune system [4].

We are here referring to pathologies mainly featured by an imbalance in the Bacteroidetes/Firmicutes ratio [5]. Some species belonging to these phyla are responsible for the production of short chain fatty acids (SCFAs), such as butyrate, propionate, and acetate [6]. The imbalance in these phyla abundance can impact gut epithelial integrity, leading to an increased permeability and undermining the immune homeostasis and the inflammatory response [7]. On the other hand, alterations in the abundance of specific microbial patterns may affect saccharolytic, proteolytic, and lipolytic metabolisms and may influence the expression of involved enzymatic pathways. However, the crosstalk interaction among all the mentioned factors has not been completely clarified yet.

Although shotgun 16S rRNA marker gene sequencing delivers interesting insights on human microbiota communities [8], it does not provide information about microbiome plasticity, especially when the adaptation to specific and mutable niche conditions is required [9]. Metaproteomics, instead, provides findings on (i) microbial constituents, (ii) the interaction between gastrointestinal (GI) microbiota and the host proteome, (iii) signal transduction, and (iv) metabolic pathways [9]. Functional shifts in microbial and human protein profiles can be further detected by using specific and curated databases allowing the identification of novel diagnostic targets and specific disease biomarkers [9,10].

Obesity and diabetes mellitus are both associated with inflammation of different tissues and organs [11]. Seeking inflammatory factors related to T1D progression, some studies highlighted findings on C-reactive protein (CRP) levels [12]. An increase in the monocyte release of interleukin (IL)-1β and superoxide radicals were also reported, suggesting an up-regulation of the inflammatory activity [13]. Besides, inflammatory processes contribute to insulin resistance in T2D. Considering that obesity is also a risk factor for the development of T2D, a large number of proteins synthesized during the inflammatory state as CRP, adipocyte-derived metabolites such as lipids, fatty acids, adipocytokines, and various inflammatory cytokines (TNF-α, IL-1β, and IL-6), have been linked to the development of insulin resistance [13,14,15].

Peripheral blood mononuclear cells (PBMC) are also involved in the crosstalk between inflammation response and dysbiosis. Pro- and anti-inflammatory activities of PBMC could be mediated by the exposure to microbial-derived SCFAs [16] or to lipopolysaccharides (LPS) coating Gram-negative bacteria [17]. Interesting insights about this mutual interaction, also in individuals affected by metabolic disease as T2D, were reported [18].

Metaproteomics allows us to build a more complete overview on protein composition at a specific time (fingerprint) and in specific health conditions, especially when used in combination with the above-mentioned meta-omics approaches [18,19].

However, metaproteomics is facing several methodological challenges due both to the ever-increasing amount of data constantly produced and the lack of standardized protocols for downstream data analysis. Advisedly, a standardized workflow is necessary to compare metaproteomics outputs belonging from different studies. This will lead to the inspection of specific associations between gut microbiota functional variations and the obesity/diabetes states. To provide a critical overview on the topic, this narrative review has been conducted considering those studies that, in the last eleven years, applied metaproteomics to investigate the onset and progression of diabetes and obesity.

Noteworthy, another milestone topic that needs to be discussed argues about the possibility to perform a pooled analysis at the individual-level included in enlarged cohorts. Recently, the Joint Programming Initiative Knowledge Platform-INtesTInalMICrobiomics (JPI KP-INTIMIC) has collected meta-data from human observational studies on gut microbiomics [20]. The human gut microbial consortium can be thus jointly analyzed in federated individual-level meta-analyses using DataSHIELD [21,22], possibly also with federated standardization of omics data [23].

## 2. Methods

### 2.1. Searching Strategy

Following the Preferred Reporting Items for Systematic Reviews and Meta-Analyses (PRISMA) 2015 guidelines [24], herein, a narrative review was based upon selective search in electronic databases (PubMed and Scholar) to identify articles in which metaproteomics was applied to investigate human and microbial metaproteome alterations in subjects affected by diabetes and obesity. Search terms included “metaproteomics”, “human”, and “gut microbiota”, along with the name of the two selected specific pathologies: “obesity” and “diabetes”. More in depth, the used searching string to query the PubMed database was “(metaproteomics OR metaproteomic OR proteomic) AND human AND (gut microbiota OR microbiome) AND (obesity OR diabetes)”. To query the Scholar database, the string “metaproteomics AND human AND gut microbiota AND obesity or diabetes” was used. The search was restricted to manuscripts written in English and published between January 2010 and February 2021. We summarized the search and selection process in the flow-diagram reported in Figure 1. After the screening phase, each paper was checked for the eligibility and inclusion criteria, consistent with the scope of this narrative review.

### 2.2. Inclusion and Exclusion Criteria

We used the PECOS format (Patients-Exposure-Control-Outcomes-Study design) to define the selection criteria of this review. Following this standard, we included studies arguing a discussion on (P) patients (men and women of all ages) affected by obesity or T1D or T2D (E) at a different state of progression, comparing their metaproteome with the healthy control group (C). Based on available information at the date of the analyses, the primary outcomes (O) were evaluated in order to assess protein variations associated with T1D, T2D, and obesity. This activity allowed us to elucidate the interactions among microbiota proteins and the relative involved metabolic pathways. Furthermore, we assessed the associations of gut microbiota functional variations in obesity and diabetes states as derived from metaproteomics alone or in combination with other omics-technologies. We here included original research (S) studies of observational design implementing a metaproteomics approach to investigate the correlation between disease status and gut microbiota. No constraint existed for the study size or the subject age and sex. Exclusion criteria included animal studies, in vitro studies, and non-original research (e.g., reviews or systematic and narrative reviews). Investigative observational studies, longitudinal or case studies, were considered, while other reviews were excluded (Figure 1).

### 2.3. Study Selection

The study selection was conducted according to a systematic search strategy: two independent reviewers screened titles and full texts of the resulting papers after following a de-duplication process. Disagreements were resolved by consultation with a third party. The selection included all studies implementing a metaproteomics approach to investigate the correlation between disease status and gut microbiota.

### 2.4. Data Extraction and Risk of Bias

The following data were extracted from each study: authors and year, cohort size and composition, subject characteristics (sex, age, and country), study design, scope of the study, omics techniques applied, and limitations (Table 1). Furthermore, the main correlations between altered proteins and pathways involved in disease status development were reported in Table 2. The included studies were critically appraised using the “Newcastle-Ottawa Quality Assessment Scale” [25]. This instrument includes three domains, namely selection, comparability, and outcomes. The article could receive one star for each item up to a maximum of nine stars and specifically: a maximum of four stars in selection, up to two stars in comparability, and up to three stars in outcomes. A high risk of bias occurred when some domains did not receive stars. In this case, the article was excluded. The NOS score for each selected study was reported in Table 1.

The evidence about GI microbial taxa diversity and metabolic diseases progressions was reported narratively due to the lack of available data from selected studies, which precluded the conduct of quantitative analyses.

## 3. Results

### 3.1. Literature Search Results and Study Characteristics

The search initially retrieved a total of 3970 records. At the end of the screening process, 10 articles were selected (Figure 1). Study baseline characteristics are shown in Table 1 while metaproteome variations are summarized in Table 2. The year of publication ranged from 2013 to 2020. Eight studies investigated human proteins from fecal samples, whereas two studies investigated human proteins harvested from plasma [26] and urinary samples [27], respectively. Four out of ten were case-control studies [27,28,29,30] while three were longitudinal studies [26,31,32]. Only three articles conducted a prospective study with a cross-sectional analysis [18,33,34]. To better highlight the findings marking T2D‘s earliest stages, the prospective study from Zhou et al., deeply profiled subjects for a long term (4 years) evaluating transcriptome, metabolome, serum cytokine levels, and proteome, as well as changes of the GI microbiome. All studies considered in this review used metaproteomics technologies based on mass spectrometry (MS). In three records, gel-free technology was combined with one-dimensional or two-dimensional electrophoresis technologies (1DE and 2DE, respectively) for protein separation [29,30,31]. With regard to the cohort size, three studies included less than 20 subjects. Among them, Ferrer et al., conducted an observational study including only two subjects, specifically one obese and one lean. Among the other seven studies, four included more than 100 individuals while three analyzed a cohort ranging between 20 and 100 individuals (Table 1). Uniquely throughout the whole dataset, Pinto et al. considered a group exclusively composed of children. Moreover, a study did not consider a group of healthy subjects as the control and instead compared results registered before and after bariatric surgery [32].

Six out of ten selected studies investigated the functional interactions between microbiota and hosts affected by T1DM [27,28,31,33] or T2DM [18,26]. The other four research articles investigated the metabolic and functional alterations in subjects affected by obesity [29,30,32,34]. Three studies out of four on T1D showed a significant alteration in host proteins of T1D patients detected in fecal samples and associated with exocrine pancreas output (CELA3A, CUZD1, α-amylase), compared to healthy subjects [28,31,33]. On the other hand, Sight et al., revealed that the urinary proteome of T1D patients compared to healthy subjects increased abundances of several lysosomal proteins (e.g., GM2A, CTSD, NAGA) associated with catabolic functions [35]. As far as it concerns studies on T2D, Zhong et al., 2019 detected a reduction in exocrine pancreas functionality, probably due to lower levels of pancreatic enzymes with respect to healthy individuals. Alterations in cytokines release was evaluated in Zhou et al., 2019, by considering differences between T2D subjects and healthy individuals. Among the papers studying obesity, only Kolmeder et al., 2015 identified and analyzed host proteins, while the others deepened only the microbial ones (Table 2).

The metaproteome quantification strategies herein adopted are based on mass spectrometry. The metaproteomics application includes the extraction and the purification phases before the MS analysis and database searching. Two different protein separation strategies were used to reduce the heterogeneous complexity characterizing different biological matrices: the gel-based methods, which include 1DE or 2DE [28,29,30], and the gel-free methods. The latter relies on chromatography techniques to separate proteins and have gradually supplanted gel-based ones. The liquid-chromatography (LC) and the high-pressure liquid-chromatography (HPLC) are faster and more convenient than electrophoresis for separating peptide mixtures [36]. Among MS analyses, ion sources of electrospray ionization (ESI), also combined with LC, and linear trap quadrupole (LTQ) Orbitrap, were adopted in four studies [28,29,31,32]. Moreover, Zhou et al., 2019 [26] used a sequential window acquisition of all theoretical fragment ion spectra-MS (SWATH MS) to evaluate the quantitative characterization of proteins (i.e., cytokines) from plasma of healthy and pre-diabetic individuals (Table 1).

### 3.2. Metaproteome Alteration in Obesity

A total of four studies assessed metaproteome variations in obesity (Table 2). Both metagenomics and metaproteomics data in Ferrer et al. revealed an increase of the relative abundances of Firmicutes and a decrease of Bacteroidetes in obese patients compared with lean individuals [29]. However, in line with Kolmeder et al., 2015, the relative amounts of expressed proteins from both phyla were very similar in obese and lean individuals. In addition, differences between the two subject groups were observed for proteins involved in cell motility, butyrate production, vitamin synthesis (B6 and B12), and starch metabolism (Table 2). The same study showed that most of the detected alterations were associated with an increased energy production by the obese gut microbiota, as indicated by butyrate production and some pili-forming proteins and flagellins that might facilitate the microbial access to carbohydrates [29]. Moreover, Kolmeder et al., 2015 reported how peptides derived from proteins involved in C5 and C6 carbohydrate metabolism, (e.g., enolase, ribulokinase, xylulokinase, phosphoketolase, and a specific glycoside hydrolase) were more abundant in non-obese individuals. Additionally, obese subjects had much more proteins involved in starch and pectin metabolism (glucosidase and pectate lyase).

On the other hand, fructose, mannose, galactose, and sucrose metabolisms resulted up-regulated in subjects belonging to the obese cohort investigated by Hernandez et al., 2013 [34]. In detail, a higher total sugar metabolism, assessed by a colorimetric assay with a set of 23 structurally diverse sugars, and a major activity of glycosidase were detected in extracted proteins from stool samples of obese individuals compared to those of lean subjects. Moreover, both Hernandez et al., 2013 and Sanchez-Carrillo et al., 2020 [32,34] highlighted a significant alteration in the expression of proteins linked to metabolic derangement, intestinal damage, and chronic inflammation state (alkaline phosphatase, serpins, and α-amylase more expressed in obese patients than healthy individuals or individual post bariatric surgery). Additionally, Sanchez-Carrillo et al., 2021 found ferritin and ferrous transport protein to be expressed in lean adults (0.46–1.0 ng/g) while both proteins resulted below the detection threshold in individuals with severe obesity.

### 3.3. Metaproteome Alteration in T1D

As above described, four studies investigated the metaproteome of subjects affected by T1D [27,28,31,33] (Table 1).

Pinto et al., 2017 found that microbial metaproteome variations of children affected by T1D were originated from Eubacterium, Faecalibacterium, and Bacteroides. The presence of these bacterial taxa is mainly linked to amino acids transport, metabolism and transcription, protein turnover, and chaperones. Specifically, the branched-chain amino acid transaminase (ilvE) and the glutamate dehydrogenase enzymes were detected among those proteins found to be more abundant in T1D subjects than healthy individuals. These proteins are involved in amino acids transport and metabolism. Additionally, regarding host proteins, T1D patients exhibited an increased expression of mucin-2 and a reduction in elastase 3A expression with respect to healthy individuals, suggesting both an increased mucin synthesis in charge of gastrointestinal epithelium protection and a reduction in exocrine pancreas functionality (Table 2).

Singh et al., 2017 considered 223 children and adolescents (age range 5–22) and observed a significant depletion of the genus Enterococcus in T1D subjects with high levels of HbA1c compared with healthy individuals. In this study, the metaproteomics was used to investigate the urinary proteome alterations in T1D subjects compared to their healthy siblings. Increased levels of lysosomal enzymes were associated with HbA1c levels. Together with these, some other proteins involved in inflammatory responses were more expressed in T1D patients. Specifically, LRG1 and CD14 resulted in the adipose tissue inflammation.

The study conducted by Gavin et al., 2018 described the alterations of both host and microbial proteins collected from children and adults affected by T1D. Five human proteins exhibited a lower level in new-onset diabetics (NODs) and seropositive individuals (positive for islet autoantibodies) compared to control subjects; the same trend was reported for three proteins secreted by the exocrine pancreas (Table 2). Moreover, two human proteins associated with inflammation, fibrillin-1, and galectin-3, were overexpressed in the T1D group. As far as concerns microbial proteins, several were differentially expressed in diseased and control individuals. The relative KEGG assignment to specific categories showed how the great part of them belongs to the phosphotransferase system, thermo-unstable elongation factors, ferredoxin hydrogenase class, and butyrate synthesis metabolism. These data revealed that proteins altered in NODs and seropositive individuals were involved in the inflammation onset, increasing the mucus secretion and the defective mucosal barrier function.

Finally, Heintz and co-workers [31], by applying a multi-omics approach to resolve the taxonomic and functional attributes of gut microbiota at the metagenomic level, found lower levels of specific human exocrine pancreatic proteins in T1D subjects (α-amylase proteins AMY2A, AMY2B, and carboxypeptidase CPA1) compared to the healthy ones. At any rate, the study was unable to identify taxa whose abundance levels correlated with those relative to pancreatic enzymes.

### 3.4. Metaproteome Alteration in T2D

Zhong et al., 2019 [18] investigated the compositional and functional changes of gut microbiota in pre-diabetic (Pre-DM), treatment naïve T2D (TN-T2D), and healthy individuals cross-sectionally to elucidate different mechanisms linked to the disease stages (Table 2). A reduction of pancreatic enzymes in Pre-DM and TN-T2D, compared to healthy individuals, was detected and implied a reduced exocrine pancreas functionality. A substantial number of Pre-DM associated microbial and human proteins were identified at the metagenomics and metaproteomics level. In fact, an enrichment in the structural domains of microbial proteins modules involved in the sugar phosphotransferase system (PTS), ATP-binding cassette (ABC) transporter of amino acids, and bacterial secretion system was detected in Pre-DM compared to normal glucose transport (NGT) individuals.

Besides, alterations in human protein production among the three groups of analyzed individuals were highlighted. The trimethylamine-N-oxide producing enzyme (FM03) was exclusively detected in the TN-T2D group. In the same group, a loss of rasGTPase-activating-like protein (IQGAP1) and unconventional myosin-Ic (MYO1C), related to the impairment of insulin signaling, were detected.

The longitudinal study by Zhou et al., 2019 [26] aimed at understanding the early disease stages of diabetes profiles by inspecting transcriptomes, metabolomes, cytokines, proteomes, and changes in the microbiome of 106 healthy subjects and individuals with pre-diabetes for four years. Correlations between microbial taxa and specific cytokines were highlighted and microbial–cytokine correlations resulted in being significant in insulin-sensitive but not in insulin-resistant participants. Barnesiella was positively associated with IL-1β (q = 0.0054), Faecalibacterium was inversely associated with TNFA (q = 0.0244), and Butyricimonas was negatively associated with four lipids only in insulin-resistant participants (q < 0.05). The study revealed that many host biochemical and microbial components are stable over time in healthy individuals even though they can undergo dynamic and marked changes in response to viral infection or other perturbations. These changes differed between insulin-sensitive and insulin-resistant individuals. By integrating information provided from proteins, cytokines, and metabolites, the pathways associated with defense responses, such as interleukin signaling pathways, mTOR signalling23, and B and T cell receptor signaling, were identified. Furthermore, during a viral infection, inflammatory pathways were differently altered in insulin-resistant participants with respect to insulin-sensitive individuals. This would suggest the presence of alterations in defense responses.

## 4. Discussion

This review was aimed at collecting the most recent scientific evidence in the field of metaproteomics to provide an overall view of the functional changes induced by interaction mechanisms occurring between the gut microbiota and host in metabolic disease conditions.

The herein selected studies highlighted proteins whose level of expression changed with respect to the health condition. These proteins were mainly involved in carbohydrate metabolism and inflammation response, but other metabolic pathways appeared also to be involved. Therefore, metaproteomics evidence may enhance the consistency of insight about host–microbiota interconnections in pathological states and may clarify how this established crosstalk can affect the inflammatory state (e.g., PBMC activation, cytokines release, inflammation serum peptide, and metabolite productions).

The great presence of bacterial strains able to ferment (metabolize) unabsorbed carbohydrates in the obese leads to an uptake of bioavailable SCFAs for the host and, therefore, an additional energy source [18,35,36]. This agrees with the currently available human case-control studies that report the association between obesity and high SCFA levels [37]. As a matter of fact, most of the metaproteome alterations were associated with the augmented energy bioavailability in the obese gut, as indicated by the increased expression of proteins involved in the refinement of carbohydrate (α-polyglucose) and starch digestion (α-glucosidase). This condition contributes to an increased SCFA production, mainly consistent in propionate and butyrate, that in obese subjects are used as an extra source of energy. Noteworthy, a controlled production of these volatile compounds in healthy subjects helps in preserving colonocyte functionality and regulates the inflammatory response [36,38]. The metaproteome of obese subjects, as reported in Kolmeder and Ferrer [29,30], is enriched in microbial proteins involved in SCFAs metabolism (e.g., butyryl-CoA dehydrogenases) compared to the metaproteome of non-obese subjects.

Ferrer et al. showed how other metaproteome variations can also affect vitamin production [29]. In this study, two microbial cobaltochelatases involved in vitamin B12 synthesis were detected in the obese gut metaproteome but not in the gut metaproteome of non-obese. This is in line with previous observations that reported higher levels of propionate in obese subjects compared to lean individuals, since propionate fermentation is mediated by a B12-dependent methylmalonyl-CoA [39]. Along with a marked metabolism of carbohydrates, an enrichment in inflammation-linked proteins (e.g., trehalase, serum C-reactive protein, and alkaline phosphatase) can be considered another discrimination element of obese metaproteome.

Ferritin and ferrous transport proteins are less expressed in obese individuals compared to lean, and this outcome is supported by studies that defined iron deficiency as a disease emerging risk factor [40]. Iron deficiency in obese subjects could be due to multiple factors that, together with a poor-quality diet, can result in a reduced iron absorption due to an increase in circulating hepcidin that, in turn, is a negative regulator of the intestinal iron absorption and the release of iron by macrophages [40].

Moreover, a substantial number of host and microbial proteins also featured the Pre-DM condition. These proteins are mainly enzymes involved in sugar transport and in the absorption from microbial cells. As with obesity, T2D is associated with an increase in carbohydrate catabolism, monosaccharide release, and with a higher amount of proteins correlated with insulin resistance (pectate lyases and sensory kinase class protein).

As a signature typical of this disease, along with hyperglycemia and insulin resistance, the T2D altered metaproteome showed a possible association between changes in the gut microbiota and low-grade inflammation (e.g., CRP synthesis).

Correlations between microbial taxa abundance and cytokine production (e.g., IL-1β, TNFA) in insulin-sensitive subjects, and the lack of these associations in insulin-resistant participants, suggest that insulin resistance may affect interactions between gut microbiome and host cytokines’ release.

This evidence is supported by the gut microbiota ability in driving stimulatory mechanisms of monocytes [11]. Moreover, inflammatory marker increases (e.g., CRP and IL-6) have been detected in apparently healthy individuals who, later, develop T2D. This suggests that inflammation occurs early, specifically during the period of impaired glucose tolerance and, therefore, prior to the diagnosis of T2D. This finding further supports the potential predictive role that metaproteomics may have in the early diagnosis of diabetes, by detecting the level of specific disease biomarkers. Undoubtedly, a better and deeper understanding is necessary on this dynamic crosstalk; therefore, the hidden mechanisms behind it need to be thoroughly investigated.

Although a negative correlation between T2D and HDL cholesterol is reported in literature [41], an unexpected and intriguing aspect emerges in T2D metaproteome. An enrichment in the ABC transporter involved in the translocation of HDL cholesterol was reported. The limited availability of information, together with the low number of studies on the T2D metaproteome investigation, suggests the need for other studies to reach a consistent view.

However, the evidence available and registered in adults and children affected by T1D showed alterations in both the host and the microbial proteins involved in specific functional categories [41], likewise those involved in pancreatic activities.

Concerning human protein alterations, the hypothesis that T1D is a combined endocrine-exocrine disease was made [42]. Exocrine dysfunction in T1D is linked to a reduction in exocrine pancreas output, as emerging from studies collected in this review that reported a decrease in the total amount of exocrine pancreas proteins both in new-onset patients and seropositive individuals affected by T1D [43,44,45]. On the other hand, in healthy subjects the expression of these proteins was positively correlated with taxa linked to gut health, suggesting a functional relationship between exocrine proteins and bacterial taxa such as Alistipes and *F. prausnitzii* [46]. The low abundance of these species in T1D individuals seems to highlight their contribution towards the gut epithelial integrity, undermining the immune homeostasis and the inflammatory response [47]. In T1D, profound metabolic changes occurred, mainly characterized by a shift from a high-level of carbohydrate metabolism to a protein metabolism. This is supported by a reduction in the expression of enzymes involved in glycolysis and, consequently, SCFA production, while human and microbial enzymes involved in proteins and amino acid metabolism increased [45,48]. Consequently, since the aromatic amino acids promote whole-body protein synthesis and inhibit protein breakdown [48], insulin-deprived T1D people show a greater abundance in circulating branched-chain amino acids (BCAA) and ketones. This could be linked to a higher expression of enzymes involved in BCAA synthesis, also in children who are T1D affected [49].

The lysosomal protein abundance in the urinary proteome of T1D patients, registered in Singh et al., 2017 [27], is in line with the increase of catabolic functions associated with T1D individuals compared to healthy subjects. Moreover, lysosomal enzymes were reported to be released from leukocytes via fusion with plasma membranes during inflammatory responses; specifically, lysosomal proteases have been implicated in diabetes associated inflammatory processes [37]. Furthermore, the disturbed lysosomal function in renal tubular cells was linked to the generation of glycotoxins (AGEs), with pathogenetic significance in diabetes [37]. These changes in lysosomal function may exert metabolic memory effects in frequent hyperglycemia [50] but, until now, no evidence has supported a possible implication in modulating the gut microbiota structure.

Certainly, the effect of diabetes and obesity on the metaproteome has been investigated in a limited number of studies and more information is necessary to evaluate the possible influence of other factors, such as dietary intake, age, or sex. As a take-home message, we reported the possibility of using a standardized metaproteomics method in clinical research. This can provide valid information about the influence of the disease on human and microbial metaproteomes. Moreover, interesting insights come from molecules that can be used as specific biomarkers to predict the onset of disease and rely on specific signatures.

Alone or in combination with other omics technologies, metaproteomics still faces methodological challenges and lacks standardization protocols. Nowadays, several approaches are available to standardize peptide identification and protein annotation processes in a way to maximize the scientific value of the obtained data and, at the same time, handle redundant protein hits [51]. Newly bioinformatic pipelines allow the reduction of protein inference issues due to the assignment of redundant proteins [52,53]. The standardization does not concern only protein annotation/identification, but also gives the chance to reduce the error frequency by standardizing the spectrum pick interpretation process. In this light, since LC-MS/MS is an increasingly powerful tool for studying proteins in the context of diseases, a rigorous standardized approach to validate individual peptide-spectrum matches (PSMs) is needed and, a method like PSM Validation with Internal Standards (P-VIS) reduces the subjectivity when evaluating the validity of PSMs [54].

Once the data are produced and made available, a standardized workflow is needed to compare metaproteomics data among different studies. The comparison is essential, especially considering an ever-increasing amount of metaproteomics findings. For this purpose, KP-INTIMIC has a huge potential in performing a pooled analysis of individual-level data following an *a priori* data standardization and harmonization. Herein, we conducted a narrative review on the associations of gut microbiota functional variations with the obesity and diabetes states, exploiting the metaproteomics approach or a combination of metaproteomics with other omics-technologies. Hitherto, to our knowledge, this is the first narrative review that merges literature results, derived from applying mass spectrometry to obtain protein profiles in diabetes and obesity.

## 5. Conclusions

To our knowledge, this is the first review offering an overview on metaproteomics studies in the low-grade inflammatory pathologies, diabetes, and obesity. Although metagenomics can provide a deep knowledge of the link between microbiota and disease state, the new metaproteomics field allows us to extend the comprehension of this complex dynamic interaction. Metaproteomics gives the opportunity of studying the functional proteomic profile under specific conditions, highlighting variations in healthy and diseased states; among these, the reduction of the activity of the exocrine pancreas in diabetics, and the alteration of iron absorption in obesity, emerged. Moreover, metaproteomics can provide information on specific biomarker expression (e.g., peptides, cytokine) that can be used as diagnostic targets in low-grade inflammation pathologies. However, the lack of comparability and heterogeneity of study designs precludes a patient or population-based conclusion. Ad hoc prepared metaproteomics databases collecting pathologic data and related metadata, together with standardized analysis protocols, are strongly required.

## Figures and Tables

**Figure 1 nutrients-14-00047-f001:**
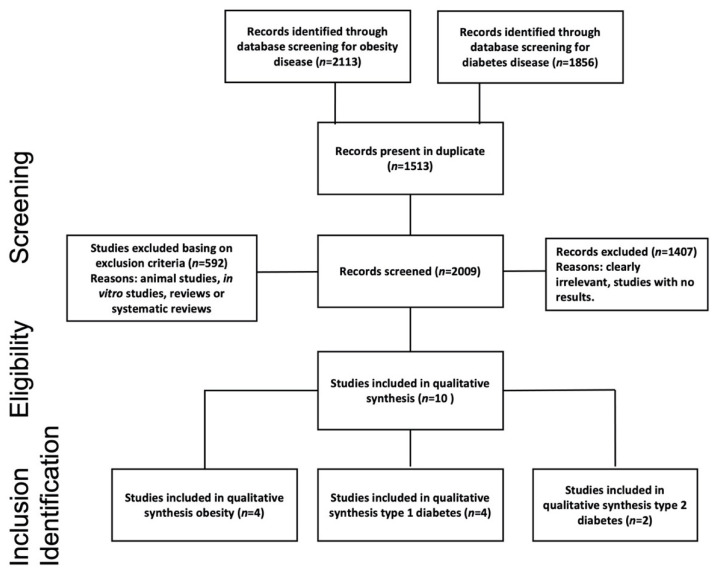
Flowchart study selection using Preferred Reporting Items for Systematic Reviews and Meta-Analyses (PRISMA) process [24].

**Table 1 nutrients-14-00047-t001:** Baseline characteristics of the included studies and Newcastle-Ottawa Quality Assessment Scale (NOS).

Authors and Year	Size Sample and Characteristics	Subjects Characteristics (Sex, Age, Country)	Scope of Study	Study Design	Metaproteomics Techniques Used	Other “Omics” Techniques Used	Limitations	NOS Score
Gavin et al., 2018	101 subjects: 33 NO, 17 SP, 29 SN, 22 CO	Denver, Colorado 46 females and 55 males Age: 9–12	Investigate functional interactions host-microbiota in subjects with T1D risk	Cross-sectional	LC-MS/MS		No information about dietary intake. Wide age range.	7
Pinto et al., 2017	6 subjects: 3 healthy and 3 T1D children	Portugal 2 males and 4 females Age: T1D children 9.3 ± 1.5 and control children 9.3 ± 0.6 years	Identify differences in the activity of intestinal microbiota between healthy and T1D children	Case-control	SDS-PAGE and LC-MS/MS (using LTQ Orbitrap)		Small number of T1D children.	6
Heintz et al., 2016	20 subjects from 4 families of at least 2 generations presenting at least 2 cases of T1D	Luxembourg 7 males 13 females Age: 5–62	Resolution of the taxonomic and functional attributes of gut microbiota and evaluation of the effect of family on gut microbiota composition	Longitudinal study (4 month)	LC (Nano-2D-UPLC-Orbirtap MS system) and MS (TopN-MS/MS method)	Metagenomics and metatranscriptomics	Need for large-scale studies.	6
Singh et al., 2017	223 subjects: 110 T1D children/adolescents and 113 healthy siblings	Washington D.C. 115 males 108 females Age: 13.9–14.5	Detection of gut microbial differences and evaluation of lysosomal dysfunctions	Case-control	LC-MS/MS		Imperfect glycemic control or subclinical inflammation in T1D patients. No information about eating habits and lifestyle.	7
Zhong et al., 2019	254 subjects: 77 TN-T2D, 80 Pre-DM, and 97 NGT	Suzhou, China 173 females 81 males Age: 41–86	Investigate compositional and functional changes of the gut microbiota to characterize different disease stages	Cross-sectional	iTRAQ-coupled- LC-MS/MS	Metagenomics	Limitations of MS-based proteomics. Confounding variables: age, drugs (CCB, hypertension, and dyslipidemia), diet, and health conditions.	7
Zhou et al., 2019	106 subjects: healthy and pre-diabetic adults	Standford, California 55 females and 51 males Age: 25–75	Understand how healthy individuals and those at risk of T2D, change over time, in response to perturbations and in relation to different molecules and microorganism	Longitudinal study (4 years)	SWATH-MS	Metagenomics, metatranscriptomics, and metabolomics	Limited studies of microbial changes. No information about diet and exercise. Heterogeneous data.	6
Ferrer et al., 2013	2 subjects: 1 lean, 1 obese	Spain 1 female (lean) and 1 male (obese) Age: 15	Identify and analyze active bacterial members and proteins expressed in lean and obese microbiota	Case-control	1D-gel electrophoresis and UPLC-LTQ Orbitrap-MS/MS	Metagenetics	No information about dietary intake.	7
Kolmeder et al., 2015	29 subjects: 9 lean, 4 overweight, 16 obese	Spain 21 females 8 males Age: 23.1 ± 2.2 (non-obese); 38.6 ± 2.4 (obese)	Characterization of non-obese and obese fecal metaproteome	Case-control	1D-gel electrophoresis RP-HPLC online coupled to MS/MS		Regular medication between obese and non-obese group.	6
Sanchez-Carrillo et al., 2020	40 severely obese adults subjected to BS	Spain Age: 47–60	Investigation the impact of bariatric surgery	Longitudinal study (3 months)	LC-ESI-MS/MS analysis	Metabolomics	Results biased for using pooling strategy.	6
Hernandez et al., 2013	13 subjects: 2 adults (β-lactam-therapy), 7 obese adolescents, 5 lean adolescents	Germany Obese: 3 females and 4 males Lean: 2 males and 3 females Age: 13–16	Evaluation of microbial shifts in relation to antibiotic treatment and obesity and measurement of carbohydrate activate enzymes	Cross-sectional	96-well plates using a BioTek Synergy HT spectrometer in a colorimetric assay		No information about dietary intake. Wide age range. Small number of subjects.	6

Abbreviations: NO, new-onset; SP, seropositive; SN, seronegative; CO, healthy control; BS, bariatric surgery; NGT, normal glucose tolerant; TN-T2D, treatment-naïve type 2 diabetic; T1D, type 1 diabetes; pre-DM, pre-diabetic; LC, liquid chromatography; LC-ESI-MS/MS, liquid chromatography-tandem mass spectrometry; LTQ, linear trap quadrupole; UPLC, ultra-performance liquid chromatography; RP-HPLC, reversed phase-high performance liquid chromatography; LTQ, linear trap quadrupole; iTRAQ, isobaric tags for relative and absolute quantification; SWATH-MS, sequential window acquisition of all theoretical mass-spectra; LC-ESI-MS/MS, liquid chromatography-electrospray ionization tandem mass spectrometry.

**Table 2 nutrients-14-00047-t002:** Summary of metaproteome variation in terms of significantly up- and down-regulated proteins in gathered/filtered studies (*n* = 10) and other metabolism pathways.

Authors and Year	Disease	Protein Origin	↑ Proteins	↓ Proteins	Metabolic Pathway/Functionality
Gavin et al., 2018	T1DM	Microbial	1. Enzymes for mucin degradation 2. Elongation factor 3. Ferredoxin reductase	4. Transferases (butyrate synthesis)	3.↑Ferredoxin catabolism 4. ↑ Butyrate anabolism
		Human	1. Galectin-3 2. Fibrillin	3. CELA-3A, 4. CUZD1 5. CLCA1 6. Neutral ceraminidase 7. IGHA1	6.↓ Sphingosine (SPH) and sphingosine 1-phosphate (S1P) 3.4.5. ↓ exocrine pancreas functionality 7.↓ IgA
Pinto et al., 2017	T1DM	Microbial	1. ilvE (BCAA transaminase) 2. Glutamate dehydrogenase (AA degradation) 3. Bifunctional GMP synthase 4. Glutamine amido transferase 5. Chaperonin GroEL	6. Phosphoketolas 7. Glyceraldehyde-3-phosphate dehydrogenase, 8. Transketolase	1.6.8. ↓ Via penthos phosphate → ↑ BCAA synthesis (Shikmic Acid Pathway) ↓ glycolysis 2.↑ NH4+ (Urea) 7. ↓ Glycolysis →↓ Piruvate ↓ SCFAs
		Human	MUC2 precursor	CELA-3A	↑ Intestinal mucin-2 ↓ Exocrine pancreas functionality
Heintz et al., 2016	T1DM	Microbial		Thiamine synthesis cofactor	↓ Thiamine synthesis
		Human		↓ AMY2A, AMY2B, CPA1, and CUDZ1	↓ Complex sugar degradation
Singh et al., 2017	T1DM	Human urinary proteome	1. LGR1 2. CD14 3. CPE 4. CTSB 5. CTSD 6. NAGA	7. Fibronectin-1 8. Pancreatic α-amylase 9. MUC1 10. PTPRN	1. ↑Inflammatory pathways (TGF-β) 3.↑AA degradation (↑urea production) 8. ↓ Exocrine pancreas functionality and ↓ complex sugar metabolism
Zhong et al., 2019	T2DM	Microbial	1. PTS 2. ABC transporter 3. FMO3 (TMAO producing enzyme)	4. Ferredoxin oxidoreductase 5. Bacterial ribosomal proteins	1.↑Phosphorylation and transport of sugar in microbial cells 2.↑ HDL cholesterol 3.↑TMAO synthesis
Zhou et al., 2019	T2M	Human	1. IL-1RA 2. CRP 3. A1C		1.↓IL-1 2.↑immune defense mechanism 3.↑ glycaemia
Ferrer et al., 2013	Obesity	Microbial	1. Glycoprotein containing FN3 2. Cobaltochelatases 3. B12-dependent methylmalonyl-CoA mutase 4. PduB 5. 3-hydroxybutyryl-CoA dehydratase 6. Butyryl-CoA dehydrogenases 7. Acetyl-CoA acetyltransferases	7. Pectate lyase 8. Aldose 1-epimerase 9. SOD 10. Pyridoxal biosynthesis lyases	1.↑ Fibrin and proteoglycans 2.3. ↑ Vitamin B12 and propionate production 4.↑ Propanediol catabolism 5. Butyrate 10. ↓ Vitamin B6
Kolmeder et al., 2015	Obesity	Microbial	1. α-glucosidase 2. Pectate lyase 3. Aminoacyl-histidine dipeptidase 4. *Bacteroidetes* proteins		1.2. ↑Starch and pectin metabolism 3. ↑AA metabolism 4. ↑SCFAs
		Human	1. Trehalase (intestinal injury and inflammation marker) 2. Alkaline phosphatase (AP) 3. Serpins (serina protease inhibitors) 4. α-amylase		1.↑ Threalosie 4.↑Starch digestion
Sanchez-Carrillo et al., 2020	Obesity	Microbial (pre-BS)	1. Enzymes involved in gluconeogenesis (glyceraldehyde 3-phosphate dehydrogenase, pyruvate orthophosphate dikinase, PEP carboxykinase, fructose-bisphosphate aldolase, glutamate dehydrogenase) 2. Enzymes involved in Acetyl-CoA synthesis (Formate C-acetyltransferase, acetyl-CoA synthase, carbon-monoxide dehydrogenase) 3. Ferredoxin oxidoreductase	4. Ferritin 5. Ferrous ion transport protein 6. Porphobilinogen synthase	1.↑Pyruvate 2.↑ Acetyl-CoA (WL pathway) 4.5. ↓Iron synthesis
		Microbial (post-BS)	1. AdhE 2. OhyA 3. SOD and perodoxins (involved in maintenance of redox balance)		1. ↑ Acetyl Acteyl-CoA → ethanol 1. Saturated fatty acid
Hernandez et al., 2013	Obesity	Microbial	1. α-polyglucose (refined carbohydrate digestion) 2. Proteins involved in pentose phosphate metabolism (PPP) 3. Proteins involved in TCA cycle		1.2. ↑ Fructose, mannose, galactose, sucrose, starch, amino sugar, and nucleotide sugar metabolism 3. ↑ Via pentose phosphate

Abbreviations: ABC, ATP binding cassette; AdhE, aldehyde-alcohol dehydrogenase; AMY2A, amylase alpha 2A; AMY2B, amylase alpha 2B; AP, alkaline phosphatase; CD14, cluster of differentiation 14; CELA3A, chymotrypsin-like elastase family member 3A; CLCA1, calcium-activated chloride channel regulator 1; CPA1, carboxypeptidase A1; CPE, carboxypeptidase E; CRP, C reactive protein; CTSB, cathepsin B; CTSD, cathepsin D; CUDZ1, CUB\zona pellucida-like domain-containing protein; FMO3, dimethylalanine monooxygenase-3; FN1, fibronectin type 1; FN3, fibronectin type 3; GMP, guanine monophosphate; HDL, high density lipoprotein; IGHA1, immunoglobulin heavy constant alpha 1; ilvE, branched-chain aminoacid transaminase; LGR1, leucine-rich-alpha-2-glycoprotein; MUC1, mucin 1; MUC2, mucin 2; NAGA, N-acetylgalactosaminidase; OhyA, oleate hydratase; PduB, 1,2-propanediol utilization protein; PEP, phosphoenolpyruvate; PPP, pentose phosphate pathway; PTPRN, receptor-type tyrosine protein phosphatase like N; PTS, sugar phosphatase system; S1P, sphingosine 1 phosphate; SCFAs, short chain fatty acids; Serpins, serine protease inhibitors; SOD, superoxide dismutase; TGF-β, transforming growth factor beta; TMAO, trimethylamine oxide; WLP, Wood-Ljungdahl pathway; A1C, glycated hemoglobin.

## Data Availability

The data presented in this study are available in the main text article.

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
