# Peer review of "Metaproteomics Approach and Pathway Modulation in Obesity and Diabetes: A Narrative Review"

_nutrients, 2021, doi:10.3390/nu14010047_

Round 1

Reviewer 1 Report

In this manuscript Calabrese et al present and discuss several experimental papers describing functional rather than structural changes on gut microbiome using metaproteomics as a research tool. This is a particularly relevant issue in the field of gut microbiome since changes on the structure of bacterial communities are quite often insignificant but the health outcomes are obvious, suggesting that either other microbes (viruses, funghi, archea) are responsible for the observed outcomes or that functional (metabolic) rather than structural alterations are ocuring among the bacterial communities.

  • The manuscript would benefit from revision of an English native speaker. Although the ideas are clear, the text can be confusing (example #1: page 1, line 43 - “gut microbiome contribute to an increased susceptibility to several disease onset and development” —> gut microbiome contribute to an increased susceptibility to the onset and development of several diseases; example #2: page 4 - “About studies on T2D, also Zhong et al. 2019 patients detected in T2D a reduction in exocrine pancreas”); example #3 (page 11): “carbohydrate metabolisms”. Sometimes, the concepts behind the general idea can be misinterpreted due to issues related with language (page 2, line 47 - “Some of these species oversee short chain fatty acids (SCFAs) production” —> some species OF THESE PHYLA oversee (?!)/are responsible for the production (?) of SCFA)
  • (page 2) - “Seeking inflammatory factors causative of T1D, some intriguing studies highlighted findings on levels of C-reactive protein (CRP)” —> how can CRP be a causative factor of T1D since the cause of this disease remais largely elusive? Also, the biological effects of superoxide are due to its radical nature rather than being an anion and thus it should be refereed as “superoxide radical”
  • (Page 2 line 67) - CRP is an acute phase protein, but is it a cytokine? Regarding the metabolites produced by the adipose tissue, it is a whopping omission not mentioning the adipocytokines…
  • (Page 3) - PECOS - please disclose the (C)omparator
  • (Page 4) - “The evidence about GI microbial taxa diversity and metabolic diseases progressions was reported narratively due to the lack of available data from selected studies, which precluded the conduct of quantitative analyses”. But did not the authors use the term “gut microbiota OR microbiome”  for literature search? How the retrieved articles do not have data on dysbiosis and diabetes or obesity (also included in the search)?
  • (Page 4) - Please explain the meaning of “MS” since it is the first time it appears in the abbreviated form… it is mentioned afterwards, in page 5. The same for DE
  • (Page 6, line 240) - The reviewer understands that this manuscript is a narrative review with the main objective of describing differences on the proteome and metabolome in metabolic disorders. However, and given the novelty of the topic, the reviewer feels that sometimes some details could be added. For instance: “T1D patients exhibited an increased expression of mucin-2 and a reduction in elastase 3A expression respect to healthy individuals” , suggesting that???)

Author Response

Here we reported the cooments for Reviewer 1, also reported in the Word file attached.

Response to Reviewer 1 Comments

Point 1: The manuscript would benefit from revision of an English native speaker. Although the ideas are clear, the text can be confusing (example #1: page 1, line 43 - “gut microbiome contribute to an increased susceptibility to several disease onset and development” —> gut microbiome contribute to an increased susceptibility to the onset and development of several diseases; example #2: page 4 - “About studies on T2D, also Zhong et al. 2019 patients detected in T2D a reduction in exocrine pancreas”); example #3 (page 11): “carbohydrate metabolisms”. Sometimes, the concepts behind the general idea can be misinterpreted due to issues related with language (page 2, line 47 - “Some of these species oversee short chain fatty acids (SCFAs) production” —> some species OF THESE PHYLA oversee (?!)/are responsible for the production (?) of SCFA)

Response 1: We thank the reviewer for detailed corrections and suggestions.

We accomplished all the raised points by extensively reviewing the whole text. Starting from the listed corrections, we modified other parts throughout the manuscript and added some details as requested. All the modified parts have been tracked in the text (review mode).

Point 2: (page 2) - “Seeking inflammatory factors causative of T1D, some intriguing studies highlighted findings on levels of C-reactive protein (CRP)” —> how can CRP be a causative factor of T1D since the cause of this disease remains largely elusive? Also, the biological effects of superoxide are due to its radical nature rather than being an anion and thus it should be refereed as “superoxide radical”

Response 2: Thanks, we modified the text at line 68 by clarifying that CRP is a protein “related to T1D”. Sorry for the mistake, we completely agree with what the reviewer pointed out: CRP cannot be defined a causative factor of T1D.

In addition, at line 70 we replaced “anion” with “radical”, referring to superoxide biological effect.

Point 3: (Page 2 line 67) - CRP is an acute phase protein, but is it a cytokine? Regarding the metabolites produced by the adipose tissue, it is a whopping omission not mentioning the adipocytokines…

Response 3: We thank the referee for have raised this omission. As correctly pointed out, CRP is not a cytokine, even though it is synthetized during inflammatory state as pro-inflammatory cytokine. We modified the text at line 80 by adding adipocytokines among the metabolites produced by the adipose tissue. The revised sentence is: “a large number of proteins synthesized during inflammatory state as CRP, and adipocyte‐derived metabolites, such as lipids, fatty acids, adipocytokines and various inflammatory cytokines (TNF-α, IL-1β, and IL-6), have been linked to the development of insulin resistance”.

Point 4: (Page 3) - PECOS - please disclose the (C)omparator

Response 4: Thank you. We addressed the raised issue by modifying the sentence at line 115 as fallow: “we included studies which (P) discussed patients (men and women with all ages) affected by obesity or T1D or T2D (E) at different progression state, comparing their metaproteome with (C) healthy group control”.

Point 5: (Page 4) - “The evidence about GI microbial taxa diversity and metabolic diseases progressions was reported narratively due to the lack of available data from selected studies, which precluded the conduct of quantitative analyses”. But did not the authors use the term “gut microbiota OR microbiome” for literature search? How the retrieved articles do not have data on dysbiosis and diabetes or obesity (also included in the search)?

Response 5: We thank the referee for raising this point. Dealing with the lack of data mentioned throughout the text, we specifically refer to the impossibility to perform a meta-analysis. Before understanding the state of the art, our first intention was to conduct a statistical comparison using metaproteomics standardized data based on stringent criteria. However, the above-mentioned aim was not feasible because of the unavailability of free raw data in single studies (in many cases both the main and the supplementary material did not contain raw data) and the heterogeneity derived by the application of different metaproteomics method approaches.

We agree with the referee that the term “quantitative analyses” is misleading, and it is not a suitable definition of what we are referring to. Thus, we modified the sentence as follows: “The evidence about GI microbial taxa diversity and metabolic diseases progressions was reported narratively due to the lack of available raw data from selected studies and their heterogeneous nature. As a matter of fact, this issue precluded the conduction of a quantitative meta-analysis.”

Point 6: (Page 4) - Please explain the meaning of “MS” since it is the first time it appears in the abbreviated form… it is mentioned afterwards, in page 5. The same for DE

Response 6: Thank you, we explained the acronyms at lines 172 and 173.

Point 7: (Page 6, line 240) - The reviewer understands that this manuscript is a narrative review with the main objective of describing differences on the proteome and metabolome in metabolic disorders. However, and given the novelty of the topic, the reviewer feels that sometimes some details could be added. For instance: “T1D patients exhibited an increased expression of mucin-2 and a reduction in elastase 3A expression respect to healthy individuals”, suggesting that???)

Response 7: Thank you, we provided further details about the main interesting differences emerged on the proteome and metabolome in metabolic disorders. The sentence at lines 254-257 reports “T1D patients exhibited an increased expression of mucin-2 and a reduction in elastase 3A expression with respect to healthy individuals, suggesting a major mucin synthesis in charge of gastrointestinal epithelium protection and a reduction in exocrine pancreas functionality, respectively (Table 2).”

Finally, in our opinion the inclusion of the graphical abstract requested by the editor improved the understanding of the whole workflow and allowed to better link the aims with the limitations we meant to describe.

Reviewer 2 Report

Review on

"Metaproteomics Approach and Pathway Modulation in Obesity and Diabetes: A Narrative Review"

by Maria Francesco Calabrese and colleagues.

In their review Calabrese and colleagues summarize microbial and human metaproteomics data in subjects affected by diabetes and obesity. The topic is very interesting as the study of the human microbiome becomes more and more attention in certain diseases. One starts to understand that a certain bacterial colonisation might be a  contributor to disease outcomes or even be a driver of disease. They tried to filter the most reliable studies and discuss the data which gives a hint on the involved metabolic processes.

Major questions:

Despite the authors put much efforts into evaluating the existing literature it was not clear to me how a study with n=2 study subjects made it on the list (Ferrer et al.), where, on the one hand, a gender difference is given (in addition the study subjects are in the puperty state!), and, on the other hand, no information on the dietary intake is given?

I was wondering that the dietary component and possible disease treatments were not dicussed in relation to the microbiome and metaproteome. To my understanding these are also important modulators of the hosts microbiome?

A suggestion would be to discuss also the techniques used to analyze the microbiome/metaproteome a little bit more as there might be differences in the resulting data, and also to discuss co-factors for which studies should be adjusted for. What would the authors think would be an optimal study design? 

Style check (Minors):

Tables: Please give tables a headline and the explanation for the abbreviations underneath the table.

When defining abbreviations in the text please stick to the same style - first write full lengt definition with the abbreviation in brackets, later on use the abbreviation.

Author Response

Here we reported the comments for Reviewer 2, also reported in the Word file attached.

Response to Reviewer 2 Comments

Point 1: Despite the authors put much efforts into evaluating the existing literature it was not clear to me how a study with n=2 study subjects made it on the list (Ferrer et al.), where, on the one hand, a gender difference is given (in addition the study subjects are in the puperty state!), and, on the other hand, no information on the dietary intake is given?

Response 1: We thank a lot the reviewer for the raised issues.

We thank her/him also for offering us the possibility to clarify that we did not consider gender and age as exclusion criteria. Our main aim was to obtain a precise study selection that would have allowed us to rely on a trusted set that can be used for discussing literature evidence on low grade inflammatory pathologies without further sub-settings.

Dealing with the studies selection, all the included studies were evaluated using the “Newcastle–Ottawa Quality Assessment Scale”. This instrument includes three domains: selection, comparability, and outcomes. An article could be excluded for high risk of bias only in the case some domains did not receive stars. The NOS score assigned to Ferrer et al., together with a good statistical analysis conducted in the study, makes it eligible for the present review (Table 1). The inclusion of this paper allowed downstream a better description of the results.

Point 2: I was wondering that the dietary component and possible disease treatments were not discussed in relation to the microbiome and metaproteome. To my understanding these are also important modulators of the hosts microbiome?

Response 2: We thank the referee to point out this issue.

We absolutely agree about the importance of discussing the capability of diet in modulating the host microbiome. However, few studies reported fragmented information about dietary intake or food habits of patients and in the majority of cases these are not exhaustive enough to formulate valid considerations.

Considering this state of the art, the evaluation of combined effect of diet and disease state on metaproteome was not possible considering the studies reported in this review.

Point 3: A suggestion would be to discuss also the techniques used to analyze the microbiome/metaproteome a little bit more as there might be differences in the resulting data, and also to discuss co-factors for which studies should be adjusted for. What would the authors think would be an optimal study design?

Response 3: We thank the referee for this suggestion.

In the paragraph 3.1 (lines 88) we highlighted that metaproteomic technics adopted in each study followed different in silico and in vitro procedural steps and, that this can largely affect the data and their subsequent standardization. In the text, we better reported that Metaproteomics “is facing several methodological challenges due to the ever-increasing amount of data constantly produced and lack of standardized protocols for downstream data analysis.” 

Hence, considering the state of the art, we proposed to always consider for an optimal study design, the use of standardized workflows (step-by-step procedural methods), in parallel with standardized downstream meta-analyses. In particular, we proposed the use of DataSHIELD to harmonize and standardized collected datasets. The need for fixing a workflow including an ad hoc experimental design is nowadays a must. Otherwise, the sensibility reached by using metaomics techniques would be lost.
